# Critical Evaluation of Various Spontaneous Polarization Models and Induced Electric Fields in III-Nitride Multi-Quantum Wells

**DOI:** 10.3390/ma14174935

**Published:** 2021-08-30

**Authors:** Ashfaq Ahmad, Pawel Strak, Kamil Koronski, Pawel Kempisty, Konrad Sakowski, Jacek Piechota, Izabella Grzegory, Aleksandra Wierzbicka, Serhii Kryvyi, Eva Monroy, Agata Kaminska, Stanislaw Krukowski

**Affiliations:** 1Institute of High Pressure Physics, Polish Academy of Sciences, Sokolowska 29/37, 01-142 Warsaw, Poland; ashfaq.ahmad@unipress.waw.pl (A.A.); strak@unipress.waw.pl (P.S.); pkempisty@unipress.waw.pl (P.K.); konrad@unipress.waw.pl (K.S.); jpa@unipress.waw.pl (J.P.); izabella@unipress.waw.pl (I.G.); kaminska@ifpan.edu.pl (A.K.); 2Institute of Physics, Polish Academy of Sciences, Aleja Lotnikow 32/46, 02-668 Warsaw, Poland; koronski@ifpan.edu.pl (K.K.); wierzbicka@ifpan.edu.pl (A.W.); kryvyi@ifpan.edu.pl (S.K.); 3Research Institute for Applied Mechanics, Kyushu University, Fukuoka 816-8580, Japan; 4Institute of Applied Mathematics and Mechanics, University of Warsaw, 02-097 Warsaw, Poland; 5CEA-IRIG-DEPHY-PHELIQS, Univ. Grenoble-Alpes, 17 av. des Martyrs, 38000 Grenoble, France; eva.monroy@cea.fr or; 6School of Exact Sciences, Faculty of Mathematics and Natural Sciences, Cardinal Stefan Wyszynski University, Dewajtis 5, 01-815 Warsaw, Poland

**Keywords:** spontaneous polarization, multi-quantum wells, nitrides

## Abstract

In this paper, ab initio calculations are used to determine polarization difference in zinc blende (ZB), hexagonal (H) and wurtzite (WZ) AlN-GaN and GaN-InN superlattices. It is shown that a polarization difference exists between WZ nitride compounds, while for H and ZB lattices the results are consistent with zero polarization difference. It is therefore proven that the difference in Berry phase spontaneous polarization for bulk nitrides (AlN, GaN and InN) obtained by Bernardini et al. and Dreyer et al. was not caused by the different reference phase. These models provided absolute values of the polarization that differed by more than one order of magnitude for the same material, but they provided similar polarization differences between binary compounds, which agree also with our ab initio calculations. In multi-quantum wells (MQWs), the electric fields are generated by the well-barrier polarization difference; hence, the calculated electric fields are similar for the three models, both for GaN/AlN and InN/GaN structures. Including piezoelectric effect, which can account for 50% of the total polarization difference, these theoretical data are in satisfactory agreement with photoluminescence measurements in GaN/AlN MQWs. Therefore, the three models considered above are equivalent in the treatment of III-nitride MQWs and can be equally used for the description of the electric properties of active layers in nitride-based optoelectronic devices.

## 1. Introduction

Spontaneous polarization has attracted considerable attention in recent decades. Despite its relatively early identification by Landau and Lifsic, the nature of this phenomenon was discussed by many authors [1,2,3,4]. These investigations include Resta’s formulation of spontaneous polarization using the Berry phase, which allowed him to calculate the difference between two different configurations of a system [5]. In addition, Resta proposed a definition of Berry phase polarization as a bulk quantity independent of the boundary conditions, which is different from the original Landau–Lifsic concept. The Berry phase formulation was used to calculate the polarization difference between two different states of the same system. If one of them is known to have zero polarization, the procedure can be used for the determination of the spontaneous polarization in the other state.

The surge of interest on polarization was driven by the emergence of applications in semiconductor devices. The focal point in semiconductor optoelectronics shifted from the vast array of semiconductors having zinc blende structure to nitride semiconductors, which have wurtzite structure. Opposite to zinc blende, the wurtzite lattice permits the existence of a vector macroscopic quantity, e.g., polarization. The effect arises from the non-equivalence between the bond along the c-axis and the c-components of the other three bonds that leads to nonzero electric dipole in the structure. As demonstrated by Karpov, the difference arises due to the deviation of crystallographic parameter u from the ideal, zinc blende equivalent value u=3/8 [6]. Therefore, spontaneous polarization in wurtzite nitrides was investigated intensively.

Polarization-induced electric fields can manifest in nanometer-size heterostructures (quantum wells, wires or dots [7,8,9,10]) and cause a shift of the transition energy between quantum states which is known as quantum-confined Stark effect (QCSE) [11,12]. In some cases, polarization has a beneficial influence on the device performance, e.g., in field effect transistors (FETs) [13,14,15]. However, in optoelectronic devices, such as light-emitting diodes (LEDs) or laser diodes (LDs), the QCSE should be reduced as much as possible. Associated with the spectral shift, there is a separation of electrons and holes within the structure, which reduces the overlap of electron/hole wavefunctions and consequently the radiative recombination rate and the emission efficiency [9,10,11,12,16,17]. The majority of nitride-based optoelectronic devices are based on polar multi-quantum wells (MQWs), with heterointerfaces perpendicular to the polarization axis, so that polarization effects are particularly strong. The attempt to obtain MQWs in different orientations was only partially successful due to the stronger lattice mismatch and the anisotropy of the surfaces [18,19].

The different III-nitride compounds have different lattice constants, so that the misfit strain is important in both heteroepitaxial layers and MQWs. The magnitude of the piezoelectric polarization can be comparable to the spontaneous polarization.

From the point of view of applications, the value of the polarization in thick layers (tens of nanometers) is not particularly relevant, since polarization-induced electric fields are screened out. However, MQWs consist of nanometer-thick layers, so the electric fields are not screened. In polar MQWs, the electric field is proportional to the polarization difference between wells and barriers. The spontaneous polarization values obtained by Bernardini et al. [20] and by Dreyer et al. [21] differ by more than one order of magnitude. The calculation method in both papers is the same, based on the Berry phase approach, but they used different reference systems. Then, in our earlier reports, we proposed a new method of calculation of the spontaneous polarization without using any reference structure [22]. Thus, three different sets of values of spontaneous polarization in III-nitrides exist. In addition, in nitrides, the polarization is strongly affected by piezoelectric effects. This creates a complex landscape of various factors that should be critically assessed.

This paper is devoted to critical assessment of the polarization properties that affect the nitride MQWs. This will be achieved by the use of the calculation scheme to determine the polarization difference between binary III-nitrides: AlN, GaN and InN. The method of determination of the technologically relevant polarization differences and the resulting electric fields in the wells and barriers will be described.

## 2. The Calculation Methods

Vienna ab initio simulation package (VASP) based on density functional theory (DFT) method of ab initio calculations was used in the calculations [23,24,25]. During initial relaxation, the atoms were moving freely. Afterwards, the structures were relaxed in a restricted way, only along the c-axis to obtain structures with a lattice constant fixed to the selected substrate. The entire movement was performed using energies obtained from the generalized gradient approximation (GGA) energy. The atoms were moved when the forces on single atoms exceeded 0.01 eV/Å. The details of the relaxation were different for different non-uniform systems. For the system presented in Figure 1, relaxation was not applied, i.e., the bulk lattice was kept rigid. For the superlattice structures, as presented in Figure 3 or Figure 7, the lattice cell vector, parallel to the c-direction, was allowed to relax freely. However, the lateral vectors were fixed to obtain structures strained to the a-lattice vectors of bulk GaN (or AlN).

As standard in VASP, the quantum states were expanded in a series of plane wave functions up to energy cutoff of 400.0 eV. Momentum space integration was replaced by summation over the Monkhorst-Pack grid (5 × 5 × 1) [26]. In order to reduce the computational size, the core functions of Ga, Al and N atoms were replaced by the projector-augmented wave (PAW) pseudopotentials adjusted to Perdew, Burke and Ernzerhof (PBE) exchange-correlation functional [27,28,29]. The only exception was gallium 3D electrons which were treated explicitly, contributing to the part of the valance band. Self-consistent (SCF) method was used to solve non-linear matrix equation with the termination energy error (residuum) set to 10^−6^ eV. The bulk lattice parameters were *a* = 3.195 Å, *c* = 5.205 Å and *a* = 3.1130 Å, *c* = 4.9816 Å for AlN and GaN, respectively. The experimental results, widely accepted in the literature, are *a* = 3.1893 Å, *c* = 5.1851 Å and *a* = 3.111 Å, *c* = 4.981 Å for GaN and AlN, respectively [30,31,32]. These results confirm good agreement between theoretical and experimental data.

PBE functional does not provide good results for electronic bandgap of semiconductors. Therefore more precise results are necessary for nitrides. Some methods are computationally demanding, including Green’s function and screened Coulomb interaction approximation (GW) [33]. Other approaches that were used in some cases are created as hybrids, including Hartree–Fock contribution in the exchange-correlation functional of Heyd, Scuseria and Ernzerhof (HSE) approach [34,35]. The latter methods could not be used in heterostructure calculations. A viable alternative was designed by Ferreira et al. [36] which is a relatively simple correction scheme called local density approximation plus half (LDA-1/2) that allows the researchers band gap energies, effective masses and band structures compatible with the experiment [37]. The LDA-1/2 approximation was not used for relaxation.

Spin-orbit contribution to system Hamiltonian was neglected. The band gap of bulk AlN and GaN were $E_{g}^{DFT}\left(\mathrm{AlN}\right)=5.95\;\mathrm{eV}$, and $E_{g}^{DFT}\left(\mathrm{GaN}\right)=3.50\;\mathrm{eV}$. Accordingly, the heavy holes (HH) and (LH) bands are degenerated at the Γ point of the valence band [38,39,40]. The only splitting of the valence band is related to crystal field; in GaN crystal, it is positive and small (ΔCR (GaN)~10 meV [41]), in AlN, it is larger and negative (ΔCR (AlN)~−165 meV [41]). Recent data suggest that this difference drastically affects emission efficiency of deep ultraviolet LEDs [42,43].

## 3. Results

The presence of spontaneous polarization can induce an electric field in bulk insulators having small size. In insulating materials, the Fermi level is located in the band gap, far away from the conduction and valence bands, and the density of mobile charge is negligible, so that the field is not screened by free carriers. In the absence of the external electric field, the electric displacement field vanishes:(1)$$D^{\left(s-pol\right)}=\varepsilon_{o}E^{\left(s-pol\right)}+P^{\left(s-pol\right)}$$

Therefore, outside of the polarized sample, the electric field vanishes, while inside, the electric field points opposite to the spontaneous polarization. Application of an external field $E_{ext}$ can be used to determine the polarization directly, when the value of the external field is tuned to obtain zero field in the sample, i.e., $E_{int}=0$. At that point, the relation between the external field and spontaneous polarization is (ϵ and $\varepsilon_{o}$ is deielectric constant and electric permittivity, respectively)
(2)$$P^{\left(pol\right)}=\varepsilon \varepsilon_{o}E_{ext},$$

Independently, the value of the spontaneous polarization of III-nitrides was determined by the Berry phase technique, which provides the polarization difference between two different phases of the same substance. A condition for the application of this method is that both the termination and the intermediate phases have to be electric insulators. A spontaneous-polarization-free reference structure is used to obtain the total value of polarization. The procedure may be applied using a different reference system, and the resulting polarization value is expected to be identical. This was questioned in the case of III-nitrides because the calculations are based on two different reference phases: zinc blende and hexagonal used by Bernardini et al. [20] and Dreyer et al. [21], respectively, obtained significantly different results. In order to compare the ab initio data, we have listed the earlier results in Table 1.

The electric fields in multi-quantum well (MQW) systems are induced by the difference of polarization between the wells and the barriers. The absolute polarization values are not relevant. In the extreme case of the identical polarization value for the wells and the barriers, the fields in the MQW vanish, whatever the polarization magnitude. In this context, the differences between polarization between AlN, GaN and InN are essential, as they govern the internal electric fields in their heterostructures.

In a multi-quantum well with a high number of periods, for the periods that are far from the system boundaries, the zero potential difference across a single well-barrier period is a good approximation. Assuming that the well (of the width *w*) and barrier (of the width *b*) have their polarizations equal to $P_{w}$ and $P_{b}$, respectively, the electric field in the well $E_{w}$ and the barrier $E_{w}$ is:(3)$$E_{w}=\frac{b\left(P_{w}-P_{b}\right)}{\varepsilon_{o}\left(w\varepsilon_{b}+b\varepsilon_{w}\right)},$$
(4)$$E_{b}=\frac{w\left(P_{b}-P_{w}\right)}{\varepsilon_{o}\left(w\varepsilon_{b}+b\varepsilon_{w}\right)},$$

Therefore, from electric fields in the wells and barriers, the polarization difference can be determined. Using a GaN-AlN well-barrier model structure, the difference can be calculated from ab initio simulations. The results of the calculations for supercells having zinc blende (ZB) and wurtzite (WZ) lattice symmetry are presented in Figure 1. In addition, the hexagonal (H) lattice was also shown. The H lattice is created by the relative translation of the metal and nitride layers along the c-axis in the wurtzite lattice to locate both layers in the same plane. Such a lattice has mirror symmetry in the c direction; therefore, the spontaneous polarization of the H lattice is zero. In order to preserve the lattice symmetry, the atoms in the supercells were located in the GaN lattice. No relaxation was allowed.

It is known that the AlN-GaN structure cannot have perfect lattice when relaxed. Therefore, the structures presented in Figure 1 were not relaxed. Generally, the polarization results from the shift of the electronic charge with respect to the positively charged atomic nuclei, leading to the emergence of an electric dipole moment. Gallium and aluminum atoms have different sizes; therefore, the wavefunction overlap between these atoms and nitrogen is different, which results in different polarization values. The calculated values of crystal orbital Hamilton population (COHP) overlaps provide a confirmation of these differences [44]. These values, obtained for GaN lattice used in the calculations, are listed in Table 2.

Note that the values for the same species, i.e., GaN and AlN, are similar for ZB and WZ, confirming well-known similarity of bonding in these structures. The hexagonal lattice is obtained from WZ lattice by the translation of the atom such that both metal and nitrogen atoms are located in the same plane. Therefore, in H lattice, the COHP values are drastically different.

For these lattices, the AlN and GaN COHP values are significantly different, indicating that the polarization-difference-related electric field should be observed in the well-barrier structures. If this is not the case, that is direct confirmation of the lattice symmetry prohibiting emergence of such a field. Therefore, H and WZ structures could serve as reference phases for Berry phase procedure and naturally, the final result, i.e., spontaneous polarization, should be the same using both references.

The band diagrams for ZB, H and WZ structures are dominated by the difference in bandgap. In addition, in the intermediate region, the electric potential curve, smoothed by the procedure described in ref. [45], has a jump of about 2 V, which is related to the existence of the dipole layer [45]. These jumps are symmetrical; thus, their total contribution to electric potential difference is close to zero. In addition, the electric sheet charge induced by polarization difference leads to emergence of electric fields in the well and barrier regions. The fields were obtained using linear approximation to potential profiles. They are listed in Table 3.

The two criteria for the polarization-difference-related field derived from Equations (3) and (4) should be fulfilled:

(i) The field in AlN and GaN should have the opposite sign;

(ii) The field in GaN should be approximately twice stronger than in AlN (due to superlattice thickness difference—24 AlN vs. 12 GaN atomic layers).

These values obtained for WZ lattice fulfilled both criteria; thus, they are a measure of the existing polarization difference. The data for ZB and H lattices are not consistent: for ZB lattice they are of the same sign, for H lattice they have a different sign but they do not obey criterion (ii). In addition, the fields obtained for ZB and H supercells are very low, respectively, one and two orders of magnitude smaller than the field obtained for WZ lattice. In summary, the results indicate that spontaneous polarization exists in WZ lattice. For ZB and H lattices, these results indicate that the polarization is negligible. Therefore, the difference in the spontaneous polarization values derived by Bernardini et al. [20] and Dreyer et al. [21] is not related to the reference phase. As these phases have zero spontaneous polarization, the Berry phase procedure using these structures as a reference should give identical results.

The values of spontaneous polarization can be used to determine polarization difference at a heterointerface as a function of the lattice parameter a. The c-axis parameter was changed according to the linear mechanical stability rule $\varepsilon_{33}=-2C_{13}\varepsilon_{11}/C_{33}$, where $\varepsilon_{11}$ is the in-plane strain and *C_ij_* are elastic constants, obtained from Mahata et al. [45]. Considering also piezoelectric effects, the polarization difference as a function of the in-plane lattice constant a is presented in Figure 2.

Despite large differences of the polarization of the particular nitrides, presented in Table 1, the polarization difference obtained using the data from refs [20,21] is similar for both the InN-GaN and the AlN-GaN systems. It is worth mentioning that the piezoelectric influence is considerable; it amounts to 50% change of the value, playing an important role in the similarity of final results for both calculations.

In order to determine the polarization difference directly, the GaN/AlN and InN/GaN MQWs were calculated by ab initio methods. First, the AlN-GaN periodic structure was calculated. The structure consists of eight atomic layers of GaN and eight atomic layers of AlN (8GaN-8AlN). The ab-initio-obtained band structure is presented in Figure 3.

The band structure presents an electric potential profile which is skewed due to the existing polarization field. This is equivalent to the presence of electric charges at both AlN-GaN interfaces. In addition, a potential jump is observed due to the presence of a polarization dipole. The obtained averaged electric potential profiles are presented in Figure 4.

The slopes of the potential profiles give information about the electric field in the structures. These fields are affected by piezoelectric effects. Therefore, the values of electric fields in the wells (GaN) and the barriers (AlN) are plotted in Figure 5.

Generally, these data provide a consistent picture. The data from Refs. [20,21] and PBE calculations recover similar trends and differ by less than 20% of their values. The data obtained using GGA-1/2 approximation have different a dependence but still remain within a 20% deviation.

A similar approach was used for the determination of GaN-InN polarization difference. Since the InN band gap is small, the number of atomic layers was reduced. The band profiles were obtained for an InN-GaN superlattice consisting of 6 InN and 12 GaN atomic layers with periodic boundary conditions. The profiles obtained for PBE and GGA-1/2 approximations are presented in Figure 6.

The PBE approximation is not adapted for the calculation of the polarization difference, as the resulting band gap is closed and the band electrons and holes screen the field, thus affecting the potential profile. The GGA-1/2 approximation still has the band gap open, so that it can be used to determine the polarization-related fields. As before, the field may be represented by the presence of electric charges at both GaN-InN interfaces. The obtained averaged electric potential profiles are presented in Figure 7.

From the slopes in the potential profiles, we can extract the electric fields in the different layers. These fields are affected by piezoelectric effects. Therefore, the values of electric fields in the wells (InN) and the barriers (GaN) as a function of the lattice parameter a are plotted in Figure 8.

These diagrams show the general agreement of the electric fields in the InN wells and GaN barriers obtained using the two polarization models in the literature, as well as direct ab initio calculations. Note that PBE calculations were affected by carrier screening, providing much lower values of the fields. The GGA-1/2 result for the smallest *a* lattice constant is approximately equal to the data from Refs [20,21]. For higher *a* values, above 3.3 Å, the electric field is reduced, due to free carrier screening. Therefore, these data should be discarded, similar to PBE results.

## 4. The Experimental Verification

The optical transition energy, measured by photoluminescence (PL) experiments is a reliable and precise method for the determination of the direct bandgap in semiconductors and semiconductor structures. Such measurements were widely applied to determine the optical transition energy. They were also used by our group in the determination of the energy of optical transitions and the direct bandgaps. Since most of the details related to the experiment were described in previous publications [12,13], we will only briefly summarize their description here.

The optical measurements were performed on a series of GaN/AlN MQW structures with equal well and barrier widths. The structures were grown by plasma-enhanced molecular beam epitaxy on 1 μm thick (0001)-oriented AlN-on-sapphire templates. The MQWs were covered by 30 nm of AlN. The well/barrier widths in the series of samples changed from 1.5 to 5 nm.

Low-temperature PL spectra were obtained, exciting the structure with several UV lines of a continuous wave Ar-ion laser (275.4, 363.8 or 302.4 nm), or using a He-Cd laser (325 nm excitation wavelength). The excitation wavelength was chosen in function of the QW thickness, and of the band-gap of the structure. The samples were placed inside a closed-circle helium refrigerator. The spectra were analyzed by a Horiba Jobin-Yvon FHR 1000 monochromator, and the signal was detected by a liquid-nitrogen-cooled charge-coupled device (CCD) camera.

The samples were characterized by high-resolution X-ray diffraction (HRXRD) using Panalytical X’Pert Pro MRD X-ray diffractometer operating at the Cu Kα_1_ wavelength, equipped with hybrid two-bounce Ge (220) monochromator, and a threefold Ge (220) analyzer in front of a detector (proportional or Pixcel). X-ray measurements confirmed good crystallographic quality of the samples, and agreement between the designed and measured well-barrier thickness period. These conclusions were supplemented by conventional and high-resolution scanning transmission electron microscopy ((S)TEM), observations that were performed in a system, TITAN CUBED 80-300 microscope, working at 300 kV and equipped with Cs-corrector and high-angle annular dark-field (HAADF) detector. These data provide information on the well and barrier thickness and chemical composition [12,13,47]. TEM analysis confirmed high quality of the samples and full agreement of their geometry with the designed structures.

XRD measurements were also used to determine lattice parameter *a* of the GaN/AlN MQWs, which was extracted from the RSMs of the (−1−124) reflection. These data were used in the ab initio simulations of GaN/AlN MQWs to obtain the energy difference between the conduction and valence band states, which are presented in Figure 9. They were compared with the PL results.

As shown, the PL energies are in good agreement with the simulation results obtained in the absence of the free carrier screening. This indicates that the system was in a semi-insulating state (SI) during measurements. The agreement is excellent for a wide range of the wells starting from 1 nm to 5 nm. The experimentally measured emission energy changes from 3.65 eV to 1.99 eV. The change is related to two factors: quantum confinement in the wells and electric-field-related QCSE. The quantum confinement accounts for the emission energy increase above bulk GaN value, i.e., $E_{g}^{exp}\left(\mathrm{GaN}\right)=3.47\;\mathrm{eV}$, whereas the QCSE is responsible for its red-shift proportionally to the QW width and the electric field value. Therefore, in the presence of electric field emission, energies decrease below bulk GaN value for wider QWs. The observed remarkable change of the emission energy down to 1.99 eV for 5 nm wide QWs is in very good agreement with theoretical data, proving excellent simulations of the electric fields by ab initio calculations.

## 5. Discussion

We present a critical comparison between various values of the spontaneous polarization and piezoelectric constants obtained from ab-initio-based analysis of the band and electric profiles in GaN/AlN and InN/GaN MQWs. These data were used to directly determine the polarization difference between the wells and the barriers.

*Ab initio* investigations were applied to determine the spontaneous polarization difference between AlN and GaN and also between GaN and InN in the three previously investigated structures: ZB, H and WZ lattices. The AlN-GaN and GaN-InN superlattices were not relaxed, having all their atoms located in a perfect GaN lattice. Thus, the piezoelectric effects affected these systems in such a way that the lattice symmetry was preserved, allowing us to study the relation between the lattice symmetry and the spontaneous polarization. It was demonstrated that the bonding in the chemically different layers was different, potentially contributing to the polarization difference. The ab initio results prove that the spontaneous polarization difference exists in WZ lattice for both system pairs, i.e., InN-GaN and GaN-AlN. In ZB and H lattice, the polarization difference was below positive identification level; thus, it may be assumed to be zero. The difference, permitted by this estimate, is one or two orders of magnitude smaller than the polarization difference in WZ lattice, showing that the spontaneous polarization difference obtained from the Berry phase procedure by Bernardini et al. [20] and Dreyer et al. [21] is not caused by the different reference phase.

The analysis of the piezoelectric properties indicates that the spontaneous polarization difference may be affected by strong piezoelectric effects amounting to a polarization change close to 50% of the spontaneous polarization value. The proper accounting of the strain caused by the a lattice parameter difference leads to the polarization difference in GaN/AlN and InN/GaN systems that is within 20% margin for Bernardini et al. [20] and Dreyer et al. [21]. In addition, the difference may have different signs for InN/GaN and GaN/AlN systems. These findings were directly confirmed by the present investigations using PBE and GGA-1/2 approximations. It was shown that the difference between different approximations may vary the obtained fields by the same amount as the difference in Refs [20,21]. Satisfactory agreement of the fields obtained directly from ab initio simulations of AlN-GaN and InN-GaN superlattices and those derived from Refs [20,21] was obtained.

These fields were compared with the results of PL measurements of optical transitions energy in GaN/AlN MQWs system grown by PA-MBE. It was shown that the PL data for all wells are in good agreement with the ab-initio-determined energies of band-to-band transitions for the a lattice constant values determined by X-ray measurements. Thus, these data were experimentally confirmed, showing that the polarization difference in nitride materials is properly determined by the data reported in Refs [20,21]. The data in Ref. [22] are also in general agreement with the PL data.

## Figures and Tables

**Figure 1 materials-14-04935-f001:**
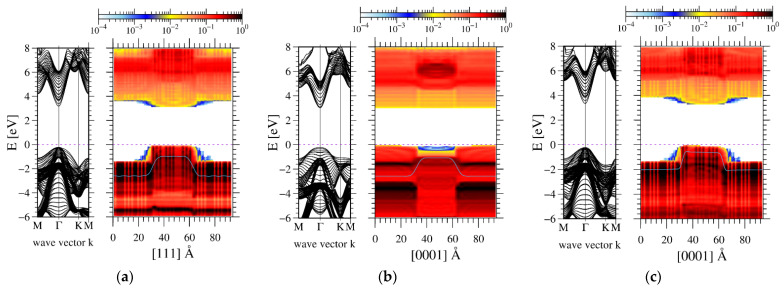
Ab-initio-calculated band profiles, obtained for 24AlN-12GaN atomic layers supercell with periodic boundary conditions: left panel—momentum space, right panel—real space. The blue line represents electric potential profile presented as electron energy: (**a**)—zinc blende, (**b**)—hexagonal, (**c**)—wurtzite structure. The GaN lattice was used in the entire system, by mere location of Ga and Al atoms in the lattice sites without relaxation.

**Figure 2 materials-14-04935-f002:**
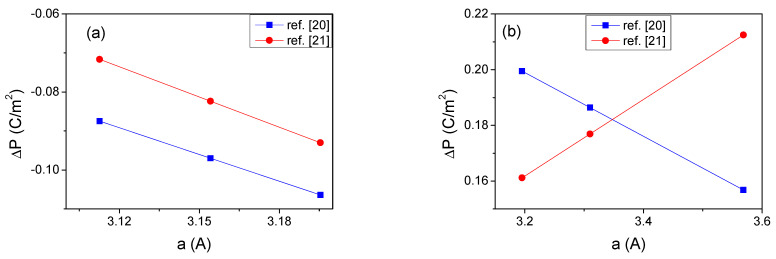
Polarization difference in function of the *a* lattice constants: (**a**) $\Delta P=P_{InN}-P_{GaN}$; (**b**) $\Delta P=P_{AlN}-P_{GaN}$.

**Figure 3 materials-14-04935-f003:**
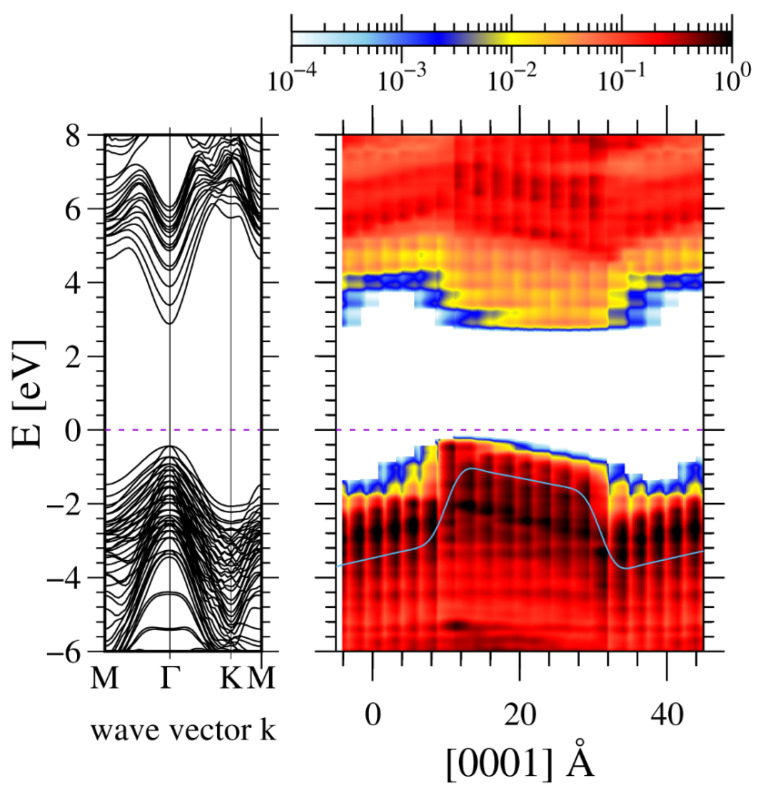
Ab-initio-calculated band profiles, obtained for 8AlN-8GaN atomic layers supercell with periodic boundary conditions: left panel—momentum space, right panel—real space. The blue line represents double averaged electric potential profile presented as electron energy [46]. GaN lattice was used in the entire system, by location of Ga and Al atoms in the lattice sites and subsequent relaxation along the c-axis.

**Figure 4 materials-14-04935-f004:**
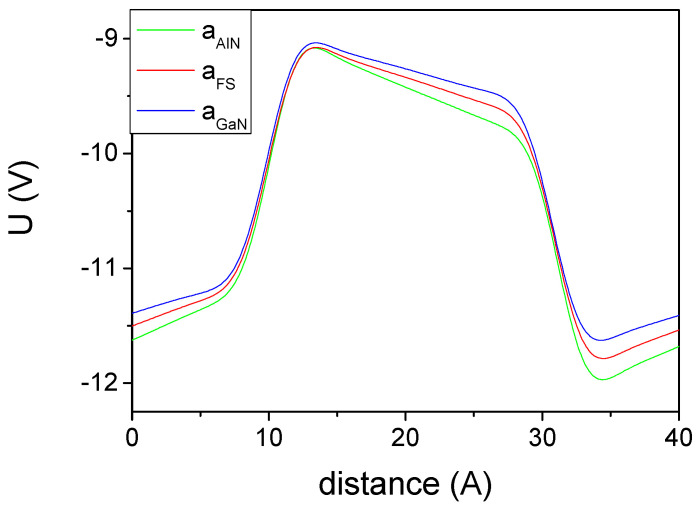
Double averaged electric potential profiles, obtained for 8AlN-8GaN atomic layers superlattice. The line represents: AlN—structure strained to AlN, a lattice constant; FS—freestanding lattice; GaN—structure strained to GaN, a lattice constant.

**Figure 5 materials-14-04935-f005:**
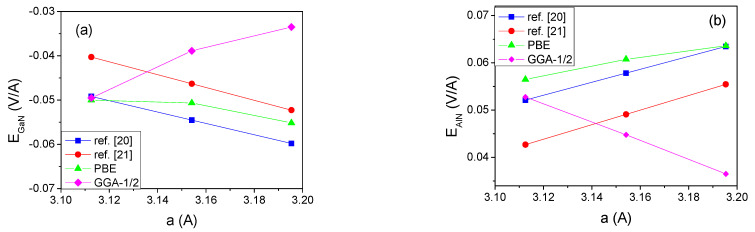
Electric fields in the well (**a**) and barrier (**b**) in 8AlN-8GaN atomic layers superlattice for various values of a lattice parameter *a*. The data denoted as refs. [20,21] were calculated using the values listed in Table 1 via Equation (3); the data denoted PBE and GGA-1/2 were obtained from ab initio electric potential using these two approximations.

**Figure 6 materials-14-04935-f006:**
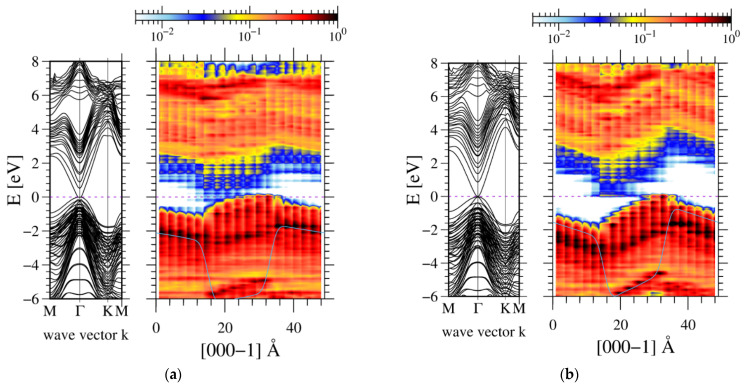
Ab-initio-calculated band profiles, 12GaN-6InN supercell with periodic boundary conditions obtained using PBE (**a**) and GGA-1/2 (**b**) approximations: left panel—momentum space, right panel—real space. The blue line represents electric potential profile presented as electron energy. Freestanding strain state was applied.

**Figure 7 materials-14-04935-f007:**
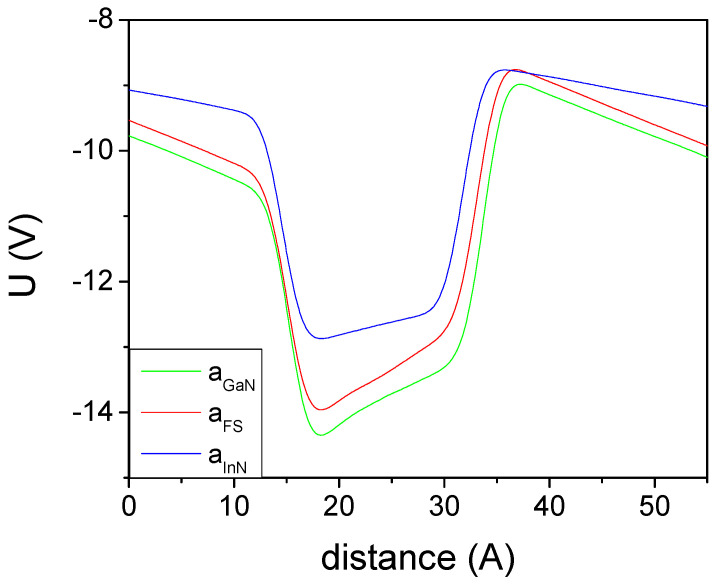
Electric potential profiles, obtained for 12GaN-6InN atomic layers superlattice using GGA-1/2 approximation. The line represents: GaN—structure strained to GaN, a lattice constant; FS—freestanding lattice; InN—structure strained to InN, a lattice constant.

**Figure 8 materials-14-04935-f008:**
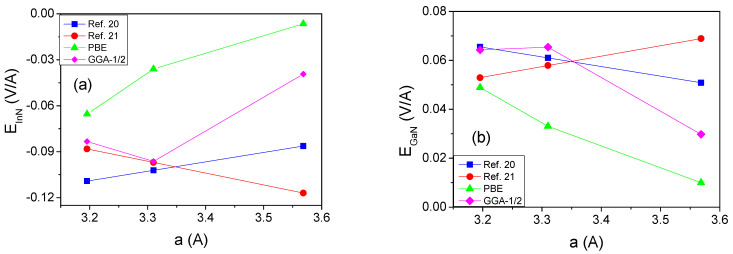
Electric fields in the well (**a**) and barrier (**b**) in 12GaN-6InN superlattice for various values of a lattice parameter *a*. The data denoted as refs. [20,21] were calculated using the values listed in Table 1 via Equation (3); the data denoted PBE and GGA-1/2 were obtained from ab-initio-derived double averaged electric potentials.

**Figure 9 materials-14-04935-f009:**
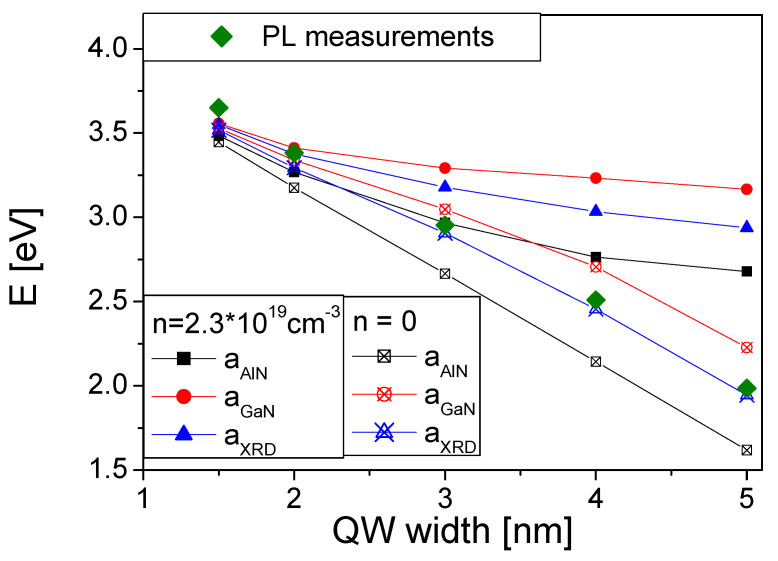
Comparison of the PL measurements of optical transition in various width GaN/AlN MQWs and the ab-initio-determined band-to-band transition energies.

**Table 1 materials-14-04935-t001:** Spontaneous polarization and piezoelectric constants published previously (in C/m^2^).

Property	References	AlN	GaN	InN
Spontaneous polarization P3	[20]	−0.081	−0.029	−0.090
	[21]	1.351	1.312	1.026
	[22]	−0.090	−0.019	−0.028
Piezo constant $\varepsilon_{31}$	[20]	−0.60	−0.49	−0.57
	[21]	−0.676	−0.551	−0.604
Piezo constant $\varepsilon_{33}$	[20]	1.46	0.73	0.97
	[21]	1.569	1.020	1.328

**Table 2 materials-14-04935-t002:** Crystal orbital Hamilton population (COHP) overlaps ΔW(in eV).

Lattice Symmetryr	AlN	GaN
ZB lattice	−16.28	−11.52
HX lattice	−3.54	−1.09
WZ lattice	17.09	−11.51

**Table 3 materials-14-04935-t003:** The electric fields in the barriers (AlN) and in the barriers (GaN) in (V/Å).

Lattice Symmetry	AlN	GaN
ZB lattice	$\left(-2.95\pm 0.57\right)\times 10^{-4}$	$\left(-5.35\pm 1.49\right)\times 10^{-4}\;$
HX lattice	$\left(1.77\pm 0.28\right)\times 10^{-5}\;$	$\left(-1.37\pm 0.39\right)\times 10^{-5}\;$
WZ lattice	$\left(1.64\pm 0.01\right)\times 10^{-3}$	$\left(-3.22\pm 0.01\right)\times 10^{-3}$

## Data Availability

The data underlying this article will be shared on reasonable request from the corresponding author.

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
