# Peer review of "Critical Evaluation of Various Spontaneous Polarization Models and Induced Electric Fields in III-Nitride Multi-Quantum Wells"

_materials, 2021, doi:10.3390/ma14174935_

Round 1

Reviewer 1 Report

Manuscript entitled “Critical evaluation of various spontaneous polarization models and induced electric fields in III-nitride multi-quantum wells” studied the polarization difference in several superlattices. It proves that the spontaneous polarization difference exists in wurtzite lattice for InN-GaN and GaN-AlN system pairs while in zinc blende and hexagonal lattice the polarization difference can be ignored, which determined that the results was not caused by the different reference phase in formers’ studies.

Reviewer evaluation:

The manuscript introduces the author’s research content and results in detail, building model structure to calculate and simulate the differences between different supercells. Compared with the experimental results, the correctness of the calculated results is proved, which is extremely rigorous and meticulous. Obviously, with further refinement, the article would be quite excellent.

Reviewer Recommendation: Minor repair

Review Suggestions:

  1. Some of the abscissa and ordinate units in the figures need to be noticed.
  2. Rows 208 to 213 would look more straightforward and beautiful as a table, as well as rows 232 to 234.
  3. The manuscript is not particularly problematic, but some minor details should be taken care of.

Author Response

Response

We agree with the Referee. Therefore we inserted two Tables (2 and 3) in the new version.

Reviewer 2 Report

This manuscript is a very good review/report where the authors compare the three models describing the role of the spontaneous polarization and piezoelectric effect on electric field distribution in different III-Nitrides MQWs. It could be published after some changes requested below.

The spontaneous polarization is a property of a bulk material, i.e. its crystal structure, which gives a certain crystallographic parameter u = (3/8)(c/a), and the ionicity of bonds, which determines the charge transferred from the anion to the cation. It is the accuracy of the prediction of these parameters that determines the accuracy of the prediction of the magnitude of spontaneous polarization. It was shown [S. Yu. Karpov, Spontaneous polarization in III-nitride materials: crystallographic revision, Phys. Status Solidi C 7, No. 7–8, 1841–1843 (2010) / DOI 10.1002/pssc.200983414. This is not my paper.] that spontaneous polarization is closely related to the value of the crystallographic parameter u. This leads to an important conclusion for ab initio calculations: if u = 3/8, as in the “ideal” wurtzite structure, then the spontaneous polarization vanishes. This is a test for the correctness of predicting the absolute value of spontaneous polarization.

Therefore, the analysis of the described work in the manuscript and comparison of theoretical predictions with Karpov’s “crystallographic” analysis is very appropriate, since the latter is based on experimental data on the parameter u and a reasonable approximation of its dependence on the c/a ratio.

This manuscript should be published after the author will add a brief discussion of Karpov’s model.

Note, that the authors use a simplified analysis of the electric field in superlattices. They do not pretend to be a detailed analysis of the band structure, taking into account the doping of individual layers, etc. In fact, they want to understand which model of polarization (spontaneous and piezo-polarization) is more adequate to the experimental situation. For such an analysis, the details of what happens inside the superlattice are not very important, since they are made up of binary materials by the authors. For this reason, it is the polarization charges at the interfaces that have the main effect on the electric field.

Author Response

Our comments are in black, , new text in the paper in blue.

Response

We fully agree with the Referee. The Karpov’s paper  explains physical origin of the electric dipole i.e. spontaneous polarization and therefore it is important in this context. We added appropriate text and the reference to the Karpov’s paper mentioned by the Referee for which we would like to thank.

Corrected text

The surge of interest on polarization was driven by the emergence of applications in semiconductor devices. The focal point in semiconductor optoelectronics shifted from the vast array of semiconductors having zinc blende structure to nitride semiconductors, which have wurtzite structure. Opposite to zinc blende, the wurtzite lattice permits the existence of a vector macroscopic quantity, e.g. polarization. The effect arises from the inequivalence between the bond along c-axis and the c-components of other three bonds that leads to nonzero electric dipole in the structure. As demonstrated by Karpov, the difference arises due to the deviation of crystallographic parameter u from the ideal, zinc blende equivalent value  [6]. Therefore, spontaneous polarization in wurtzite nitrides was investigated intensively.